# Delay-Based True Random Number Generator in Sub-Nanomillimeter IoT Devices

**Maulana Randa [1,2,\*]** , **Mohammad Samie [1]** and **Ian K. Jennions [1]**

1   Integrated Vehicle Health Management Centre, Cranfield University, College Road, Bedford MK43 0AL, UK; m.samie@cranfield.ac.uk (M.S.); i.jennions@cranfield.ac.uk (I.K.J.)
2   Research and Development Centre, Indonesia Ministry of Defense, Jl. Jati No.1, Jakarta 12450, Indonesia
\*   Correspondence: maulana.randa@cranfield.ac.uk

**Abstract:** True Random Number Generators (TRNGs) use physical phenomenon as their source of randomness. In electronics, one of the most popular structures to build a TRNG is constructed based on the circuits that form propagation delays, such as a ring oscillator, shift register, and routing paths. This type of TRNG has been well-researched within the current technology of electronics. However, in the future, where electronics will use sub-nano millimeter (nm) technology, the components become smaller and work on near-threshold voltage (NTV). This condition has an effect on the timing-critical circuit, as the distribution of the process variation becomes non-gaussian. Therefore, there is an urge to assess the behavior of the current delay-based TRNG system in sub-nm technology. In this paper, a model of TRNG implementation in sub-nm technology was created through the use of a specific Look-Up Table (LUT) in the Field-Programmable Gate Array (FPGA), known as SRL16E. The characterization of the TRNG was presented and it shows a promising result, in that the delay-based TRNG will work properly, with some constraints in sub-nm technology.

**Keywords:** true random number generator; IoT; near-threshold voltage

## 1. Introduction

In the era of the internet of things (IoT), everyone feels the need for privacy and security because their private data is floating around in the connected cloud [1]. As IoT-based systems have both hardware and software requirements, there is always a potential for systems to be hacked, if the hardware is not as well-secured to a suitable level as the software. Research in recent years has demonstrated the existence of malware that could be removed from the system, if appropriate software-level countermeasures are set up properly [2]. Hackers might target such malware for hacking the physical systems. Therefore, hardware security is also an essential requirement, in addition to the security of the software, to ensure that the security of the system and the privacy of the user's data are well-established [2].

Cryptography is now essential for securing access to both the data and hardware, which is necessary for IoT-based systems [3]. A key is an important aspect in cryptography, and it can be created using a random number generator (RNG). There are two types of random number generator; a true random number generator (TRNG) and a pseudo-random number generator (PRNG). The comparison between TRNG and PRNG has been summarized in Table 1.

**Table 1.** Comparison between a true random number generator TRNG and a pseudo-random number generator (PRNG).

|                      | TRNG                | PRNG                       |
| -------------------- | ------------------- | -------------------------- |
| Source of randomness | Physical phenomenon | Mathematical algorithm     |
| Uniformity           | Yes                 | Yes                        |
| Independence         | Yes                 | No (Periodic/deterministic) |
| Efficiency           | low                 | high                       |

While a pseudo-RNG (PRNG) is simple to implement and sufficient enough for many applications, there is always a desire to have a TRNG, especially for highly critical systems. The reason for this is that PRNG was created from a computational algorithm that has deterministic properties. When the algorithm behind the PRNG is compromised, the random number that it generates is also compromised. On the other hand, a TRNG utilizes a physical system that has intrinsic randomness, which can be extracted to create an RNG. This results in non-deterministic properties for the TRNG.

There have been various designs and technologies suggested for architecting TRNGs for different types of IoT. For a big-sized IoT, such as a smart-fridge, smart-toaster etc., a TRNG that uses optical scattering [3,4] and radioactive decay [5] as its source of randomness (SoR) can be used. While these TRNG are bulky and have low efficiency, they have a good randomness property. For a smaller IoT device, such as a smartphone, the use of sensors, such as an accelerometer and gyroscope, as the source of randomness has been reported to have excellent results [6–8]. However, the implementation of these approaches still relies on external data processing, e.g., a PC, which is impossible to include in resource-constrained devices, such as an implanted IoT like a pacemaker. For this type of IoT, a TRNG that utilizes the intrinsic parameters of the devices is preferred, as they do not have to rely on the external source of randomness.

An all-digital RNG implemented in 65 nm and 14 nm technology was proposed in [3,4], respectively. Pamula [4] proposes a high-quality RNG, based on a processed low-quality RNG with intrinsic SoR. Their analysis shows excellent performance. However, the technology used is too big for an implanted IoT. In [3], the author implements TRNG in the latest semi-conductor technology. However, the source of randomness used is not always available in the IoT devices, making it difficult to achieve in IoT. In modern Field-Programmable Gate Array (FPGA) technology, the SRL16E is standard and it has the potential to be used in TRNG. The author in [9,10] uses the SRL16E, and configures it to be a ring counter, in order for it to become one of the components of their TRNG. However, they only use the ring counter as a complementary component to increase the periodicity of the RNG, and not as the primary source of randomness.

Moreover, the size of transistors in the future will become smaller beyond nano-millimeter technology [5]. This causes the electronic devices to run at a near-threshold voltage (NTV) [6]. These phenomena have an impact on the critical timing of the device, because the distribution of the process variation is non-gaussian [7]. A couple of research studies have been done to address this issue [8,11]. However, from the extensive literature review, a report on the effect of NTV in the time-critical application, such as a delay-based random number generator, is not in existence.

This paper presents a study on the implementation of delay-based TRNG, intending to explore TRNG performance and properties in sub-nm technology. The sub-nm delay-based RNG was modeled in the FPGA, using a ring counter based on the SRL16E configuration of Xilinx's Look-Up Table (LUT) as the main source of randomness.

The rest of the paper is organized as follows: Section 2 provides an introduction to RNG implementation in the FPGA and the metrics for RNG characterization. The experimental setup, practical limitations, and a framework for location selection are presented in Section 3. The results, findings, and statistical analysis are discussed in Section 4. Finally, the paper will be concluded in Section 5.

## 2. Related Works

### 2.1. Random Number Generator

The idea of a random number generator is based on stochastic modeling, in which an observable random variable can be obtained from a random phenomenon. In a random number generator, let $S$ be the state space of the generator, which is also a subset of a set $\Omega$. The random variable generated is part of the random space $U$ that is extracted using the extraction function $g$. What is being extracted by $g$ is a mapped state space S by function $f$ so that $f:S \rightarrow S$.

$$
\begin{aligned}
\Omega &\in S \\
f &: S \rightarrow S \\
g &: S \rightarrow U
\end{aligned}
\tag{1}
$$

In a True Random Number Generator (TRNG), $f$ is the physical source of randomness and $g$ is the logic or function used to process the source of randomness further. In a Pseudo-Random Number Generator (PRNG), $f$ and $g$ are the mathematical algorithms used to generate the random number. Both TRNG and PRNG need an initial condition. In TRNG, the initial condition is any current state of the physical system, while in PRNG, the initial condition needs to be provided by the seed $x_0$. Figure 1 is given to illustrate this mechanism.

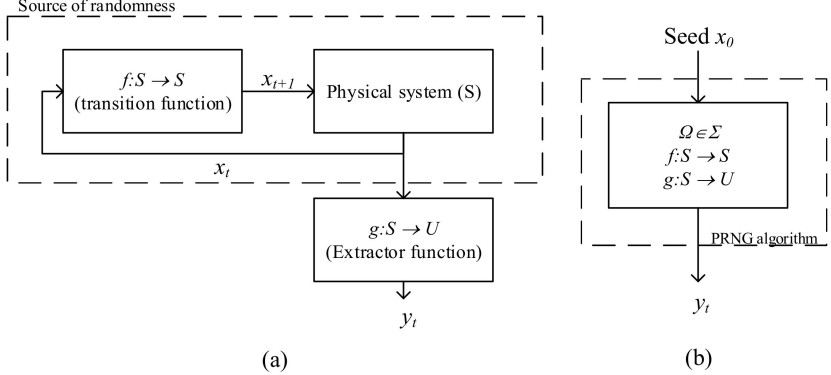

**Figure 1.** Random number generator (RNG) configuration (**a**) TRNG and (**b**) PRNG.

In semi-conductor devices, one of the sources of randomness is from the shift in the D.C. current, also known as the burst noise [12]. This phenomenon happens because of the modulation of the current flowing over a physical barrier. The magnitude of this current can be calculated using the Schottky equation, as in Equation (2).

$$
\overline{i^2} = 2qI_D\Delta f
\tag{2}
$$

where $q$ is the electronic charge, $I_D$ is the average value of the random current pulses at the drain of the transistor, and $\Delta f$ is the measurement bandwidth. From Equation (2), it can be seen that the bigger the bandwidth of the measurement, the higher the current will become. It also suggests that the higher the pulsating current at the drain, the more that the noise will increase. Burst noise is mostly caused by a random variation such as crystallographic defects in the bipolar junction transistor. Impurities can slip into the defect during the manufacturing process and form a low resistance current path. When the current flows over this resistance, some of it will leak, meaning that the output has current inconsistency.

Another source of randomness in semiconductor devices that comes from a random variation in the manufacturing process is the flicker noise. Flicker noise is also known as $1/f$ noise, because it mainly affects the lower frequency range, i.e., the Megahertz frequency. Two theories can be used to explain the flicker noise phenomenon, namely the number fluctuation theory [13] and mobility fluctuation theory [14]. Number fluctuation theory explains that flicker noise happens because there is

an inconsistency in the number of electrons that can pass through the defective current path at any given time. Counter to this, mobility fluctuation theory states that flicker noise does not have any correlation with the number of electrons that pass the defective current path. The velocity inconsistency of that electrons causes this. However, both theories agree that the main cause of the flicker noise comes from the defective current path of the transistor which is a random variation of the manufacturing process.

### 2.2. Random Number Generator in FPGA

FPGA refers to embedded electronics comprised of a vast number of digital circuitries, known as primitives, that can be employed to configure a wide range of different applications. There are two methods of RNG integration for FPGA applications, either through building mathematics models of RNGs using FPGA primitives, which results in PRNGs [15], or by utilizing the random variation of the FPGA manufacturing process, thus creating TRNGs [15]. Although the FPGA provides a sufficient level of randomness with a high throughput for PRNG applications, the random number that it generates is no longer secure if the function behind the PRNG is compromised. This rest of this section focuses on the implementation of TRNG in the FPGA.

There are two main components used to build the TRNG in the FPGA: the source of randomness and the extractor. First, the source of randomness is the combination of the state space $S$ and transition function $f$, as in Equation (1). The transition function $f$ is to prepare the state space $S$ for generating the next random number.

An example of a digital circuit that can be used as a source of randomness for TRNG when implemented in the FPGA is the shift-register. The shift-register consists of a chain of flip-flops. With the FPGA from Xilinx, the shift-register can be simplified by utilizing the SRL16E [16], which is a particular mode of the LUT in Xilinx's FPGA. This configuration will significantly reduce silicon area usage.

The second component to build TRNG in the FPGA is the extractor. One of the easiest ways to create an extractor function in the FPGA is by using a comparator circuit to compare the quality of the two sources of randomness. If one source of randomness is better than the other, it will generate bit "1", and if it is the other way around, it will produce bit "0".

A simplified block diagram of TRNG for the implementation in the FPGA is presented in Figure 2. The oscillation frequency of the two Sources of Randomness (SoR) will be counted by the binary counter and compared in a comparator circuit to generate the 1-bit random number. The 1-bit random number will be stored in a register and concatenated. After the first random number generation is finished, the finite state machine will tell the counter to start the SoR frequency counting routine again. In order to generate a 128-bit random number, it needs to run 128 times.

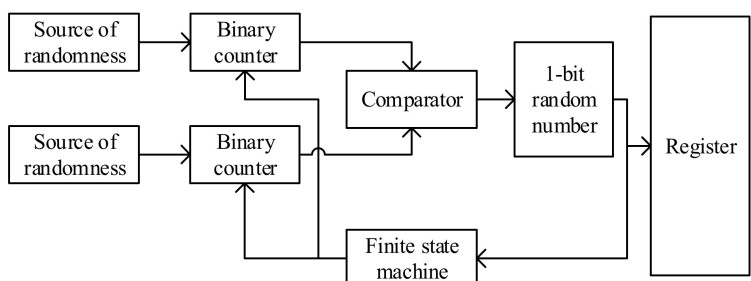

**Figure 2.** Block diagram of TRNG implementation.

### 2.3. Test for Randomness Analysis

There are a couple of concepts that have been proposed in the literature to measure the quality of the bits produced by the random number generator. One of the parameters that can be easily tested with a non-statistical method is the frequency test. The frequency test aims to measure the uniformity of the bits produced by the RNG. An ideal uniform binary random number contains the same number

of ones and zeros. This means that it has 50% uniformity while 100% uniformity means that the random number is made up of all ones or all zeros.

Another concept to measure the quality of a random number generator is by using a statistical test. There are a couple of suites used to test the randomness of an RNG such as DieHARD [17], FIPS140-2 [18], AIS-31 [19], NIST SP800-22 [20], and TestU01 [21]. Table 2 shows the differences between each test suite.

**Table 2.** Minimum Input for Different RNG Test Suites.

| Name of the Test Suite | No. of Test Types | Minimum Input (bit) | Year of Publication |
|---|---|---|---|
| DieHARD | 18 | 2.5 million | 1995 |
| FIPS 140-2 | 11 | 16 | 2001 |
| AIS-31 | 9 | 3 million | 2001 |
| TestU01 | 266 | 32 (max) | 2007 |
| NIST SP800-22 | 15 | 1 million | 2010 |

George Marsaglia published dieHARD in 1995 as an improvement of a random number of quality measurement techniques developed by Donald Knuth. His idea was to fix the *p*-value to a pre-chosen interval [$\alpha$, $1 - \alpha$]. Beforehand, the *p*-value is not fixed, which makes it challenging to interpret the result of the randomness test.

The Federal Information Processing Standard (FIPS) 140-2 is a standard created by the National Institute of Standard and Technologies (NIST) in 2001. They also created a new standard called the NIST SP800-22. This is the latest tool used to quantify the quality of a random number generator. The difference between the two is that FIPS is more a qualitative way to standardize a random number generator. In contrast, NIST 800-22 is a more quantitative way to measure a random number generator. However, FIPS140-2 has been criticized by industries, because it takes too long to get a random number certified. The certification process cannot be done by the creator of the random number themselves—it must be done by a third-party company.

AIS-31 is an improvised version of FIPS 140-1. It also introduces a new testing technique that focuses on how to measure the quality of a random number that has been post-processed. The tests that are included from FIPS 140-1 are the mono bit test, the poker test, the run test and the most extended run test. The other test used is the autocorrelation test, the uniform distribution test (which includes two sub-tests), a comparative test for a multinomial test and the last one is the entropy test.

TestU01 is considered to be the most comprehensive test as it combines 266 test suites from the existing test suite available in the literature and commercial products. It divides the test into three packs: (1) "Small Crush", which consists of 10 tests; (2) "Crush", with 96 tests; and (3) "Big Crush", which consists of 160 tests. However, it only able to handle 32-bit inputs, which are too limited for modern RNGs in a cryptographic application.

In this paper, NIST SP 800-22 Rev. 1a will be used. It is a current standard that is widely used and accepted to measure the randomness of the random number generator. It consists of 15 statistical tests as described in Table 3. Every test has several parameters such as minimum bit length (*n*), block length (*m* or *M*), and several sub-tests. The number of *n* needs to be supplied by the user while m and M are parameters that can be set within the test suite.

NIST SP800-22 is widely used in industry and commercial RNG products because it is considered as having a low tolerance to error. As a result of this, it is hard to pass the NIST SP800-22 unless the RNG is perfect. This claim is confirmed by [22], which mentions that high numbers of a good RNG have difficulty passing 20% of the NIST test.

Every random number test suite measures the *p*-value of the RNG. The *p*-value refers to the probability that the RNG under test will have the same quality as the referenced RNG used in the test suite. The *p*-value is chosen to represent the quality of an RNG to understand whether an RNG is

good or bad. Nevertheless, it does not provide any information on which part of the RNG makes it a lousy RNG.

**Table 3.** Statistical Tests Within National Institute of Standard and Technologies (NIST) SP 800-22.

| Test Name | n | m or M |
|---|---|---|
| Frequency Test | $n \geq 100$ | - |
| Frequency Test within a Block | $n \geq 100$ | $20 \leq M \leq n/100$ |
| Runs Test | $n \geq 100$ | - |
| Longest-Run-of-Ones | $n \geq 128$ | - |
| Binary Matrix Rank | $n \geq 38{,}912$ | - |
| FFT | $n \geq 1000$ | - |
| Non-overlapping Template | $n \geq 8m - 8$ | $2 \leq m \leq 21$ |
| Overlapping Template | $n \geq 10^6$ | - |
| Maurer's Universal Statistical | $n \geq 387{,}840$ | - |
| Linear Complexity | $n \geq 10^6$ | $500 \leq M \leq 5000$ |
| Serial Test | | $2 < m < [\log 2\, n] - 2$ |
| Approximate Entropy | | $m < [\log 2\, n] - 5$ |
| Cumulative Sums | 100 | - |
| Random Excursions | $n \geq 10^6$ | - |
| Random Excursions Variant | $n \geq 10^6$ | - |

The *p*-value is compared to a significance level $\alpha$, which is set by the tester. If the *p*-value is lower than $\alpha$, then it means that the RNG is rejected as being a good RNG. NIST recommends setting the value of $\alpha$ to 1%. This means that there is a 1% probability that the RNG under test will be as good as the referenced RNG. However, as every RNG test suite is a statistical test, there are two types of error. Type I, also known as a false-positive error, happens when the test suite fails to detect a lower *p*-value of the RNG under test when it has a small *p*-value.

On the other hand, type II, also known as a false negative error, happens when the test suite fails to detect a higher *p*-value of RNG under-test when it has a high *p*-value. According to [22], a small *p*-value does not mean that the RNG is terrible. Instead, it tells us that there is a high chance of type II error, which is more important from a practical point of view.

*2.4. Metrics*

Cryptography applications need a high rate of random number generation. The parameter used to measure the rate of the random number generation is known as the throughput. Throughput is calculated using Equation (3).

$$throughput = \frac{n \times f_{max}}{latency} \tag{3}$$

where *n* is the number of bit-length of the generated random number, $f_{max}$ is the maximum working frequency of the design, and *latency* is the number of cycles taken to generate the 1-bit of random number. $f_{max}$ is obtained by looking at the post-route-and-placement report of the FPGA and not the maximum frequency of the FPGA board.

**3. Experimentation**

*3.1. Design of Ring Counter RNG (RCRNG)*

The RNG based on the ring counter circuit will be implemented in the Kintex-7 FPGA development board, which consists of 7K325T FPGA from Xilinx. It utilizes 28 nm technology, which is still widely used in critical systems nowadays, such as in avionics and radar technology. The Kintex-7 FPGA is categorized as a −2 L device, which means that it has a nominal voltage of 0.9 V. It is understandable that the threshold voltage for 28 nm devices is 0.4 V, and the experiment should ideally run at that voltage level. However, this experiment uses the nominal voltage of the FPGA, because lowering the

voltage beyond the recommended voltage can harm the FPGA. While the differences between the nominal voltage used and the threshold voltage is an interesting topic to discuss, this paper is focused on the effect of the non-uniform distribution of the process variation caused by the near-threshold voltage to the delay-based TRNG. This will leave the research about the environmental effects, such as voltage and temperature differences, to others.

In this paper, SRL16E will be used as the primary source of randomness for TRNG. The motivation behind it is to test the feasibility of upcoming silicon technology, where the size of the transistor will become smaller. As stated in [16], LUT in SRL16E mode has very short wiring so the delay should be small enough to affect the timing or power consumption. This property will be used as the model for future delay-based TRNGs in FPGAs, where the wiring is tiny. However, the differences in the delay are too small to be measured with today's technology. Therefore, in this experiment, the configuration of the shift-register from SRL16E for creating the ring counter is used. A ring counter is a shift-register with a feedback loop. The introduction of the loop will increase the delay to the measurable value of today's measurement technology.

The main component to build the RC-based TRNG is sliceM. It contains LUTs that can be programmed as a 16-bit shift-register in the form of SRL16E from the UNISIM library. By instantiating the LUT as shift-register, the resource usage of the FPGA can be minimized.

The idea of using a ring counter as a source of randomness for TRNG is the same as the idea of using a ring oscillator to create a delay of a system clock. Two ring counters initialized as 10101010 . . . or 01010101010 . . . will oscillate when activated. Depending on the process variation of the components used to create the ring counter, the oscillation frequency will be different from one ring counter to another. A 1-bit random number can be generated by comparing the frequency of two ring counters. In this experiment, the 16-bit ring counter was initialized to only have one bit of 1 and 15 bits of 0. This configuration was used to create a more significant delay, so then the signal analyzer can easily see the difference in frequency. However, this configuration will increase the latency of the design and affect the overall throughput.

The TRNG is built based on the block diagram shown in Figure 2 without the finite state machine. This configuration is then stacked in parallel, as in Figure 3. The reason for this is that by using a parallel configuration, it is possible to generate an n-bit of random numbers in one run. This configuration also increases the confidence level of the measurement and the bit generation because it minimizes the effect of temperature and voltage change.

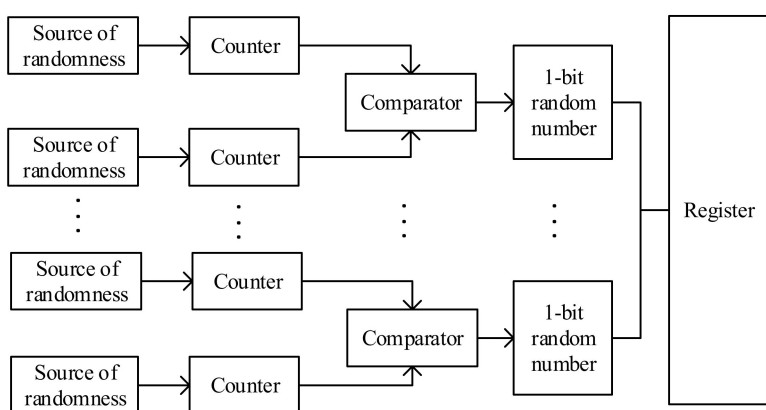

**Figure 3.** Block diagram of TRNG implementation.

## 3.2. Experimental Limitation

Before implementing the RCRNG, there are a couple of things that need to be considered. The first is to find the location on the FPGA floorplan where the pair of RCs that will be compared can produce the best entropy for the random number generator. This can be done by inspecting every possible location in the FPGA floorplan. After that, every likely pair of RC also needs to be checked to find the

best entropy. However, there are 16,000 sliceMs in Kintex-7 7K325T FPGA. A single sliceM consists of 4 LUT that can be programmed as a four 16-bit shift register (SRL16E). Therefore, there are $2^4 = 16$ possible combinations on a single sliceM. Testing all of the possible combinations of all of the potential locations means testing 256,000 possible combinations, which will be time-consuming. Therefore, some constraint needs to be applied to the experiment by limiting the number of RC pairs that will be tested as follows:

1.  The test will only be done by comparing the neighboring LUTs on the same slice. This makes it only possible to compare two pairs of LUT per sliceM.
2.  To acquire the data, an integrated logic analyzer, in this case, Chipscope Pro 14.7, was used. Even though it is a powerful tool to debug the circuit design of the FPGA, there are some practical limitations. For the Kinetix-7 FPGA, the maximum number of signals that it can read at a single time is 4096. Hence, in order to test all of the possible pairs by applying the constraint on point (1), the measurement needs to be done (16 × 16,000)/4096 times, or about 62 times, which is a time-consuming process. For this reason, the test will be limited to as close as to the maximum number of signals of Chipscope as possible, which is 4000 RC pair. Each pair will be captured 1000 times, in order to be able to understand the uniformity of the ring counter pair.
3.  The process of placement will be done manually by applying the location constraint to the ring counter pair, and the relative location constraint to the counter so then it is located close to the ring counter.
4.  Even though there are some limitations in this experiment, it still gives a clear idea about the steps needed to find the best location for the RC pair to generate a random number with the best entropy. The flowchart in Figure 4 is given for a better understanding of the location selection process.

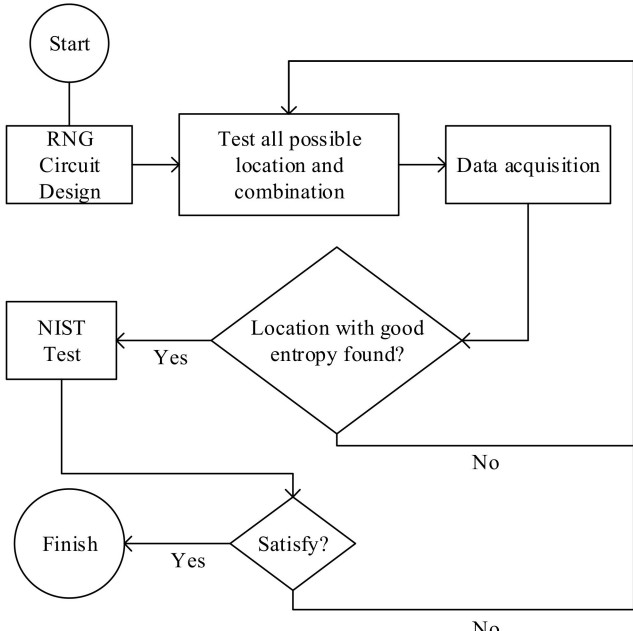

**Figure 4.** Location selection flowchart.

Secondly, concerning the more technical aspect of the design, the delay in every part of the circuit needs to be the same up to the counter logic. In the FPGA, the delay on the ring counter is not a problem, and it is assumed to be the same. This is because in the FPGA, the ring counter was made by instantiating a LUT, which means that no wiring is needed to connect the component that builds the ring counter. However, it is a bit of a challenge to make sure that the delay between the ring counter and the counter is the same. This is because the manual routing tool from ISE is complicated to use. Therefore, in this experiment, the delay from the ring counter to the counter is made as small and as

similar as possible by forcing the placement of the counter to be as close as possible relative to the ring counter circuit.

Lastly, it is desirable to create a hard macro of the RCRNG circuit (at least from ring counter to counter circuit) to fix the location, to lessen any delays between the components and to make sure that there is no additional logic inserted into the circuit. However, it remains a big challenge for FPGA designers to create a hard macro from an instance that has an initialization value in one of the components of the hard macro. In this case, the ring counter circuit needs to have an initial value which will have consequences on the presence of a power net. In ISE 14.7, the tool does not accept any power nets inside a hard macro. In this experiment, to make sure that there is no additional logic added to the path between the ring counter and the counter, they need to be forcibly located as close as possible, relative to the ring counter circuit. This can be done by using `rloc` constraint.

## 4. Findings and Analysis

From the 4000 pairs of the ring counter, the uniformity of the bits generated from each RC pair can be calculated. In Figure 5, the RC pair that have 100% uniformity are not shown to clarify the graph. Perfect uniformity in a bit string is reached when the number of ones is the same as the number of zeros, indicating that the uniformity is 50%. From Figure 5, there are 45 RC pairs that have precisely 50% uniformity. However, the initial design was to create a 128-bit random number. Therefore, another run of tests is needed to confirm that the RC pairs that have 50% uniformity. After undertaking the process another two times, 32 and 55 RC pairs were found after the second and third location finding process, as shown in Figures 6 and 7. In total, 132 locations with 50% of uniformity were found, which is sufficient to build the 128-bit RNG.

NIST SP800-22 was used to measure the quality of the random number. Some of the tests in the NIST test suite need at least $10^6$ bits of data, so at least 10,000 bitstreams are needed for a 128-bit RNG. This will translate into sequence length (128) and bitstreams (10,000) for the input of the test suite. Raw data from the pre-selected RC pair is fed into the test suite, and the results are as shown in Table 4. The first ten columns are ten bins from 0 to 1. What is in the bin is the *p*-value that falls within the range of that bin. For example, 3024 in the first row and first column means that there is a 3024 *p*-value, which has a value between 0 to 0.1 in the frequency test. For each test, the optimum result is achieved when the *p*-value is distributed uniformly across all bins. The *p*-value column is the uniformity of the *p*-value. The $\chi^2$ test determines the uniformity of the *p*-value. The optimum value for the uniformity of the *p*-value is 1. However, according to the guideline of the NIST test suite, it mentions that the minimum value of 0.01 for the uniformity of the *p*-value is enough for the RNG under-test to pass each test.

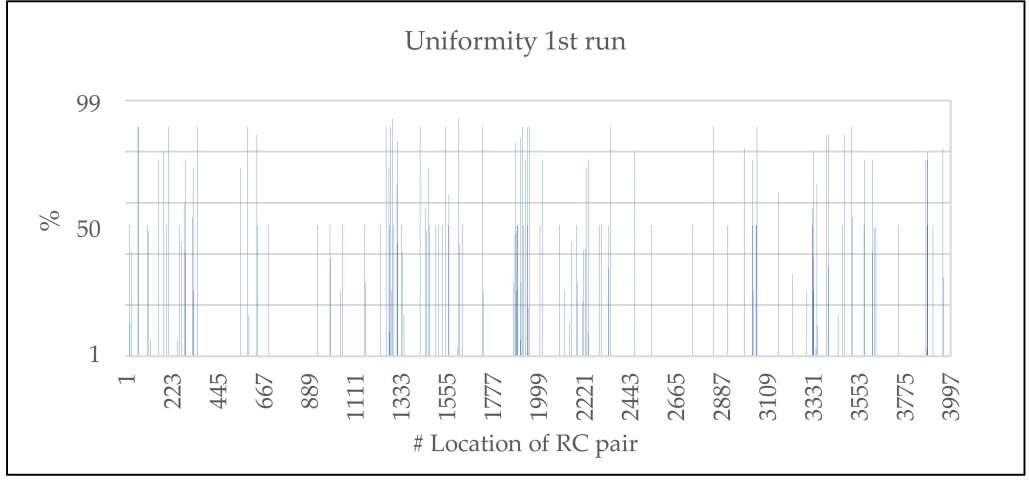

**Figure 5.** Uniformity of the RC pairs on first run.

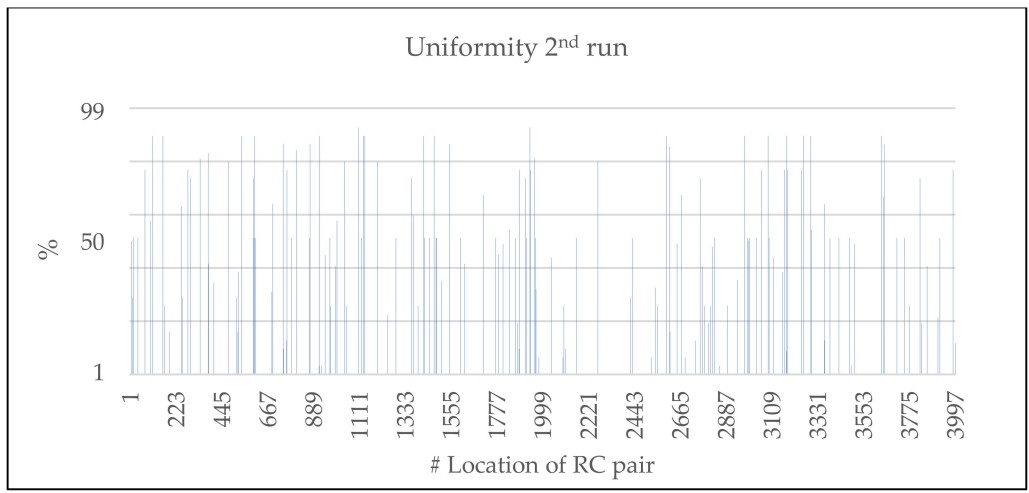

**Figure 6.** Uniformity of the RC pairs on the second run.

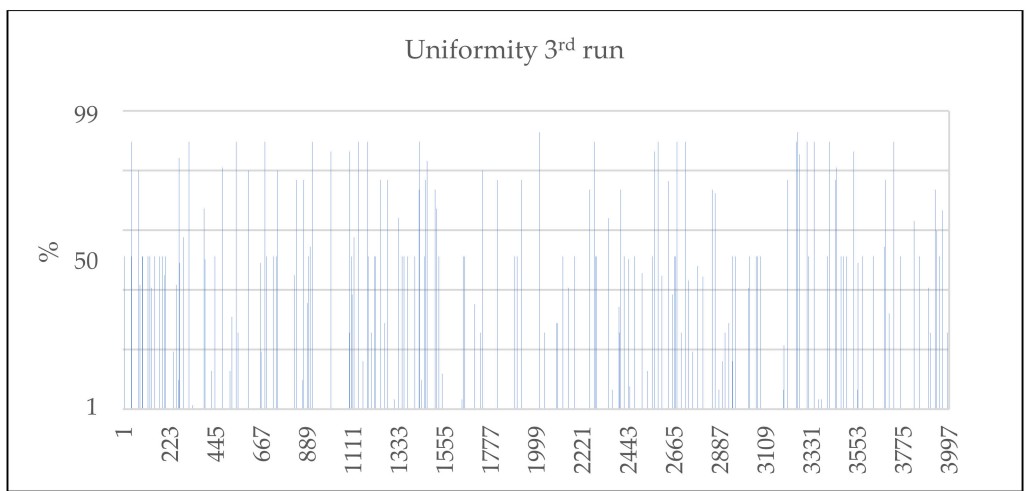

**Figure 7.** Uniformity of the RC pairs on the third run.

**Table 4.** NIST SP 800-22 Test Results.

| C1 | C2 | C3 | C4 | C5 | C6 | C7 | C8 | C9 | C10 | *p*-Value | Statistical Test |
|---|---|---|---|---|---|---|---|---|---|---|---|
| 3024 | 1067 | 1395 | 728 | 780 | 841 | 0 | 850 | 846 | 469 | 0 | Frequency |
| 3024 | 1067 | 1395 | 728 | 780 | 841 | 0 | 850 | 846 | 469 | 0 | Block Frequency |
| 2903 | 1247 | 782 | 898 | 948 | 586 | 554 | 541 | 925 | 616 | 0 | Cumulative Sums |
| 1092 | 1019 | 1104 | 1041 | 905 | 1070 | 764 | 1046 | 1091 | 868 | 0 | Runs |
| 2208 | 1227 | 1013 | 995 | 827 | 1030 | 610 | 800 | 646 | 644 | 0 | Longest Run |
| 10,000 | 0 | 0 | 0 | 0 | 0 | 0 | 0 | 0 | 0 | 0 | Rank |
| 1156 | 1292 | 0 | 1987 | 0 | 2566 | 0 | 0 | 2999 | 0 | 0 | FFT |
| 1503 | 0 | 0 | 0 | 0 | 0 | 0 | 0 | 0 | 8497 | 0 | Nonoverlapping Temp |
| 10,000 | 0 | 0 | 0 | 0 | 0 | 0 | 0 | 0 | 0 | 0 | Overlapping Template |
| 0 | 0 | 0 | 0 | 0 | 0 | 0 | 0 | 0 | 0 | 0 | Universal |
| 0 | 0 | 0 | 0 | 0 | 0 | 0 | 0 | 0 | 10,000 | 0 | Approximate Entropy |
| 0 | 0 | 0 | 0 | 0 | 0 | 0 | 0 | 0 | 0 | 0 | Random Excursions |
| 0 | 0 | 0 | 0 | 0 | 0 | 0 | 0 | 0 | 0 | 0 | Rand Excursions Var |
| 1380 | 0 | 0 | 0 | 7654 | 0 | 0 | 0 | 0 | 966 | 0 | Serial |
| 10,000 | 0 | 0 | 0 | 0 | 0 | 0 | 0 | 0 | 0 | 0 | Linear Complexity |

The NIST test, however, has a rigorous rule where the recommended significance level is between 0.1%–1%. This means that it will only tolerate an error of 1%. For example, if the number of *p*-values that falls within a bin is outside of the range of ±1%, then it will be considered an error, and will fail the $\chi^2$ test. As an example, in this experiment, 10,000 bitstreams were produced. If the bitstreams

are divided into ten bins, each bin should have a 1000 ± 1% *p*-value fall into it. According to Table 4, none of the bins satisfy this rule. Therefore, when the program calculates the uniformity of the *p*-value, it will give error igamc: UNDERFLOW. This means that the calculated uniformity of the *p*-value is too small. This is the reason why the values of the *p*-value column are all zeros.

Aside from using the $\chi^2$ test, [20] also suggests another way to analyze the uniformity of the *p*-value, which is using a graphical plot of the *p*-value. Figure 8 has been given to further analyze the results of the NIST. Figure 8 is the representation of Table 4 in graph format. To clarify the graph, the maximum range of the y-axis is limited to 1500. In this graph, most of the *p*-values fall outside the 1000 ± 1% range, but not by much. The only bin that falls way over the tolerance range is C1. This result is an indication of a type II error, where most of the *p*-value falls into the low bin. By looking at the proportion of sequences that pass the test, even though it falls below the tolerance range of 9900/10,000, the result is not bad at all. As already discussed in [22], even the build-in PRNG of NIST SP800-22 has only a 15% probability of passing all tests. Therefore, it can be said that the result of this experiment does not mean that the RNG fails to produce an excellent random number, but rather a type II errors in the statistical measurement, which is sometimes more useful from a practical point of view.

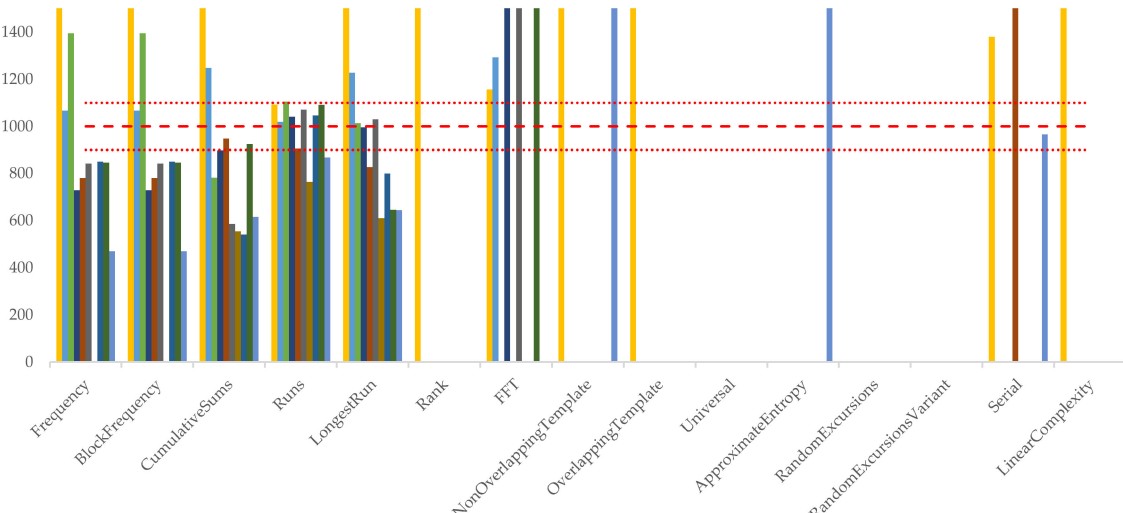

**Figure 8.** Graphical presentation of the NIST test result.

The results from Table 4 leads to the suspicion that something is not right with the NIST test suite because of the parametrical error. Therefore, a test using the NIST built-in PRNG was done to verify that the NIST test suite is working as intended. In this case, the built-in PRNG that was used is the linear congruential generator. The test was run using 1 million, 10 million, 100 million and 1000 million bits, to see the effect of the number of input bits on the results of the test. The results are shown in Figure 9. When the NIST test suite is fed with the minimum input recommended by the standard, it did not return any meaningful data. The uniformity is not valid using the minimum input, even though the generator under test is from the built-in PRNG. When the number of input bits increased, the uniformity of the *p*-value improves. When 100 million and 1000 million input bits were given, the test result returned the same *p*-value and the same uniformity of *p*-value as well. From this test, it can be concluded that, in order to get a meaningful result from the NIST test, a more significant number of bits is needed than the recommended minimum input bit mentioned on the standard.

Based on this finding, another NIST test was conducted using the experimental data. This time, 10 million bits were used as the input of the test suite, and this number was increased to 100 million to see the effect of increasing the number of input bits, and how it relates to the output obtained from the NIST test. First, 10 million bits was divided into 1000 sequences with a length of 10,000 each. The result can be seen in Table 5. It shows that there is an improvement in the uniformity of the *p*-value, as expected. However, when the number of input bits was increased to 100 million, the test returned

an error message saying that the number of bits is insufficient. The same error message also reported by [23]. Nevertheless, the result from Tables 4 and 5 agree with the trend in Figure 9. This means that, despite the inability to acquire the results for the NIST test with a higher input bit, the RCRNG can pass the NIST test when it is tested with a larger input bit.

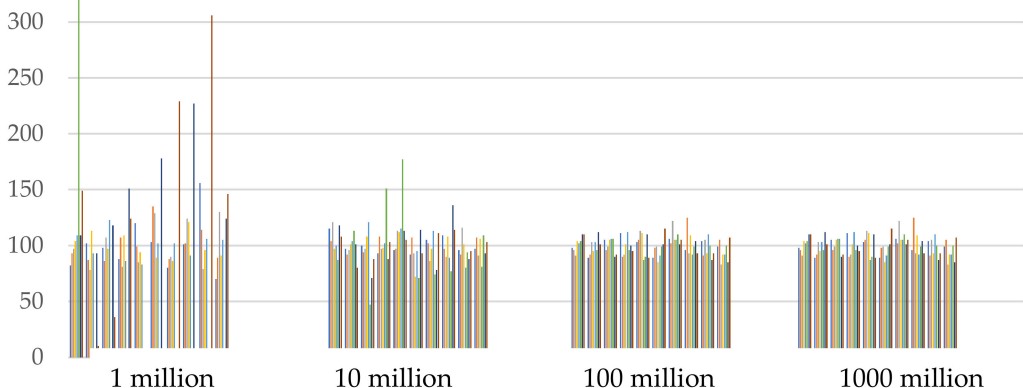

**Figure 9.** Comparison of the different input bit lengths in the NIST test.

**Table 5.** NIST test results with 10 million input bits.

| C1 | C2 | C3 | C4 | C5 | C6 | C7 | C8 | C9 | C10 | *p*-Value | Statistical Test |
|---|---|---|---|---|---|---|---|---|---|---|---|
| 535 | 82 | 83 | 55 | 44 | 41 | 54 | 39 | 30 | 37 | 0 | Frequency |
| 1000 | 0 | 0 | 0 | 0 | 0 | 0 | 0 | 0 | 0 | 0 | Block Frequency |
| 772 | 105 | 57 | 31 | 18 | 9 | 7 | 0 | 1 | 0 | 0 | Cumulative Sums |
| 922 | 18 | 8 | 9 | 12 | 8 | 7 | 2 | 7 | 7 | 0 | Runs |
| 706 | 111 | 60 | 38 | 28 | 24 | 8 | 11 | 6 | 8 | 0 | Longest Run |
| 104 | 150 | 78 | 103 | 172 | 54 | 66 | 86 | 96 | 91 | 0 | Rank |
| 614 | 124 | 46 | 46 | 33 | 31 | 30 | 34 | 15 | 27 | 0 | FFT |
| 102 | 85 | 95 | 123 | 91 | 94 | 124 | 96 | 105 | 85 | 0.042531 | Nonoverlapping Template |
| 611 | 165 | 30 | 45 | 51 | 32 | 25 | 15 | 16 | 10 | 0 | Overlapping Template |
| 0 | 0 | 0 | 0 | 0 | 0 | 0 | 0 | 0 | 0 | 0 | Universal |
| 1000 | 0 | 0 | 0 | 0 | 0 | 0 | 0 | 0 | 0 | 0 | Approximate Entropy |
| 0 | 0 | 0 | 0 | 0 | 0 | 0 | 0 | 0 | 0 | 0 | Random Excursions |
| 0 | 0 | 0 | 0 | 0 | 0 | 0 | 0 | 0 | 0 | 0 | Random Excursions Variant |
| 999 | 1 | 0 | 0 | 0 | 0 | 0 | 0 | 0 | 0 | 0 | Serial |
| 116 | 72 | 100 | 83 | 102 | 119 | 100 | 100 | 112 | 96 | 0.029401 | Linear Complexity |

Another discovery from the test result as shown in Tables 4 and 5 is the distribution of the *p*-value from the experimental data that gravitates towards the smaller *p*-value (column C1). This can be interpreted as one indication of small periodicity. The reason for this phenomenon might come from the non-gaussian distribution of the process variation of the NTV devices. It can be concluded that, when the SRL16E was used as a model source of randomness for RNG in sub-nanomillimeter electronics, the RNG can still perform well but with small periodicity.

Using the XPower Analyzer tool from Xilinx, the estimated power consumption is 0.157 Watts. Table 6 presents the resource utilization and throughput of the proposed design compared to the other TRNG implementation in the FPGA. Based on the post Placement and Route (PAR) analysis, the maximum frequency for this design is 74 MHz. In this experiment, the latency is $16 \times 16$, because of the initialization of the ring counter as mentioned in Section 3.1. Therefore, the throughput of the design is calculated as 37 Mbps using Equation (1). From Table 6, it appears that the RC-based TRNG has the right balance between FPGA resource utilization and its throughput.

**Table 6.** Throughput comparison between the TRNG implementation in the Field-Programmable Gate Array (FPGA).

| RNG Type | Resource Utilization (LUT) | Throughput (Mbps) |
|---|---|---|
| PLL [24] | 6144 | 69 |
| Ring Oscillator [25] | 3968 | 13.8 |
| Metastability [26] | 8960 | 50 |
| Chaotic Oscillator [27] | 43,732 | 58.76 |
| Ring Oscillator [28] | 7296 | 4.77 |
| Ring Counter | 2048 | 37 |

The throughput can be increased by increasing the number of ones at the initialization stage of the ring counter. It can be increased up to 0.6 Gbps, when all of the bits on the ring counter are initiated as ones. However, there is a drawback to this. The faster the ring counter overflows the frequency counter, the harder it is for the comparator to see any differences in frequency. It will think that the frequency of the two ring counters are the same and it will generate the same bit every time. For the application of a random number generator, this property is unwanted. However, for the application of a physical unclonable function, this configuration will create a more stable bit generation, which is preferred by many researchers.

## 5. Conclusions

In this paper, a random number generator based on the ring counter circuit was implemented in the FPGA. The framework for the construction process was described, as well as the process of location selection, to get the best randomness out of the ring counter pair. Because of the limitation of the IDE tools used, there are a couple of practical limitations in the experiments. This limitation has been explained thoroughly and overcome. The evaluation using the NIST SP800-22 statistical suite was also presented, and the results have been discussed thoroughly. One comment for the NIST test suite is that one needs to have a significant input a bit beyond its recommended minimum input to get a meaningful result.

In terms of the adaptability of delay-based RNG for sub-nano millimeter technology, it was shown that the current delay-based RNG can still be implemented. Even though the path delay is small and negligible, there are still some differences in delay or frequency that can be extracted to construct a random number generator. However, one should take note that the periodicity of delay-based RNG in the sub-nano millimeter will be small.

**Author Contributions:** Conceptualization, M.R.; Data curation, M.R.; Formal analysis, M.R.; Investigation, M.R.; Methodology, M.R.; Resources, M.R.; Software, M.R.; Supervision, M.S. and I.K.J.; Validation, M.R. and M.S.; Visualization, M.R.; Writing—original draft, M.R.; Writing—review & editing, M.S. All authors have read and agreed to the published version of the manuscript.

**Funding:** This research received no external funding.

**Conflicts of Interest:** The authors declare there to be no conflict of interest. The funders had no role in the design of the study, in the collection, analyses or interpretation of data, in the writing of the manuscript or in the decision to publish the results.

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
