# Peer review of "Delay-Based True Random Number Generator in Sub-Nanomillimeter IoT Devices"

_electronics, doi:10.3390/electronics9050817_

Round 1

Reviewer 1 Report

This paper describes a delay-based true random number generator in the FPGA. Here are some suggestions to improve the quality of the manuscript:

  1. It would be helpful if the authors could report the estimated power consumption and occupied area because it is already an FPGA implementation and the area efficiency is also quite important for TRNG.
  2. A figure of the experiment setup would be better to describe the experimental results more clearly.
  3. It would be beneficial to the readers if the authors could include a comparison table to show the performance and advantages of presented techniques.
  4. Sub-nm devices do not necessarily operate at a near-threshold voltage unless the power supply voltage is close to the threshold voltages of the devices. However, the FPGA that the authors used in the article is implemented in a 28 nm CMOS process. It means the nominal power supply voltage is around 1.1V but the threshold voltages of the devices are around 0.4V, which is far from the near threshold voltage operation. Please explain what power supply voltages that the authors used and their impacts on the presented techniques.

Reviewer 2 Report

This paper is about delay-based true random number generator. The paper implemented RC-based TRNG. Overall, the experiment is well conducted. However, the paper only implemented previous RC-based TRNG on FPGA. There is no novel idea discovered by author. The author should provide new approach and method. Afterward, the method is compared with the state-of-art for fair comparison.

Round 2

Reviewer 2 Report

The author clearly mentioned the novelty in the response. The overall paper is well-written. For this reason, I recommend to accept this paper.

Author Response

Thank you for your recommendation. We also have revised the manuscript and perform proofreading by a certified proofreader to improve grammar and writing. 

Reviewer 3 Report

The authors have made significant attempts to improve their work based on the reviewer's comments.

It now appears that the work done is prudent (to the best of this reviewer's knowledge).  Unfortunately, the presentation of the work is poor due to insufficient English grammar and writing style.  Beginning from the first sentence of the abstract there are grammatical errors.

"True Random Number Generator (TRNG) uses the physical system as their source of randomness."

should be...

"True Random Number Generators (TRNGs) use the physical systems phenomenon as their source of randomness."

This is only an example.  The majority of the writing needs to be revised in a similar manner.  I recommend the authors revise the work with very careful attention to proper grammar, and writing.

Author Response

Thank you for your recommendation. We have revised the manuscript and perform proofreading by a certified proofreader to improve grammar and writing.  

Round 3

Reviewer 3 Report

The authors have thoroughly proof-read their work and made many minor edits to improve the grammar throughout.  The manuscript now reads easily, which makes the work appear much much better.